# Effect of Exogenous Plant Growth Regulators and Rejuvenation Measures on the Endogenous Hormone and Enzyme Activity Responses of *Acer mono* Maxim in Cuttage Rooting

**DOI:** 10.3390/ijms241511883

**Published:** 2023-07-25

**Authors:** Xinxin Zhou, Ruyue Li, Hailong Shen, Ling Yang

**Affiliations:** 1State Key Laboratory of Tree Genetics and Breeding, School of Forestry, Northeast Forestry University, Harbin 150040, China; zhouxinxin41@163.com (X.Z.); li5833654@163.com (R.L.); 2State Forestry and Grassland Administration Engineering Technology Research Center of Korean Pine, Harbin 150040, China; 3State Forestry and Grassland Administration Engineering Technology Research Center of Native Tree Species in Northeast China, Harbin 150040, China

**Keywords:** *Acer mono* Maxim, twig cuttings, adventitious roots, endogenous hormones, oxidases

## Abstract

The cuttage rooting method for *Acer* species is difficult to achieve a good efficacy as trees maintain good characteristics at the rejuvenation stage, thus improving the rooting of *Acer* species. The addition of exogenous hormones and rejuvenation can improve the rooting effect of cuttings; however, the specific regulatory mechanism is still unclear. Here, *Acer mono* Maxim rejuvenation and non-rejuvenation cuttings were used as test subjects, to investigate the effects of exogenous hormones on the activities of endogenous hormones and antioxidant enzymes in the rooting process of young cuttings. The results showed that exogenous growth-regulating substances significantly improved the rooting rate of *A. mono*. Exogenous hormones naphthylacetic acid (NAA) + indolebutyric acid (IBA) increased the initial levels of the endogenous hormones, indoleacetic acid (IAA) and abscisic acid (ABA), and the enzyme activities of peroxidase (POD) and polyphenol oxidase (PPO). Rejuvenation treatment prolonged the time of increase in ABA content and indoleacetic acid oxidase (IAAO) activity at the root primordium induction stage, while increasing trans-zeatin riboside (ZR) content and decreasing POD enzyme activity in cuttings. These results demonstrate that *A. mono* cuttings can achieve the purpose of improving the rooting rate by adding the exogenous hormone (NAA + IBA), which is closely related to the changes of endogenous hormone content and enzyme activity, and these changes of *A. mono* rejuvenation cuttings are different from non-rejuvenation cuttings.

## 1. Introduction

Trees have vigorous growth and rooting abilities and a strong stress resistance at the rejuvenation stage, which gradually weakens with increasing age. Rejuvenation refers to the complete or partial reversal of plant characteristics at maturity during the entire developmental process, regressing to a previous stage of juvenile development and regaining some or all plant growth characteristics during infancy [1]. Therefore, it is of great significance to maintain the trees’ excellent characteristics at the rejuvenation age, using rejuvenation measures, such as cutting off, promoting budding, spraying hormones, etc.

*Acer mono* Maxim, also known as Pentagonal Maple, belongs to the *Acer* genus of Aceraceae. It is a perennial tree or shrub with a unique tree shape, leaf shape, leaf color, and forest appearance and important economic and ecological values. It is widely distributed in Northeast China and scattered throughout South China [2]. Cao experimented on the seeds of purple maple in the Zhangguangcailing area and found that only 20% of the autumn leaves of the progeny were still purplish red at the seedling stage, losing the good characteristics of the maternal plant [3]. To preserve the good characteristics of *A. mono*’s maternal plants, cutting propagation, with simple operation and low cost, was a better choice. However, *A. mono* is a difficult species to root from cuttings. Studies found that auxins significantly improve the AR initiation of tea cuttings [4]. Zhou found that 1500 mg/L of naphthylacetic acid (NAA) + indolebutyric acid (IBA) (1:1) increased the rooting rate of *A. mono* shoots to 66.67% [5]. Endogenous hormones play an important role in the regulation of rooting from cuttings. It is known that the endogenous hormone, indoleacetic acid (IAA), plays an important role in the induction and expression of adventitious roots and callus formation. Abscisic acid (ABA) is a stress hormone. A low content of ABA in cuttings is conducive to callus formation and induces root primordia to form adventitious roots. It also promotes starch hydrolysis to form sugars in cuttings, providing energy for adventitious root growth [6]. The endogenous hormone, gibberellic acid (GA_3_), plays a synergistic role in sensing injury signals and improving resistance and is believed to inhibit the formation of adventitious roots in cuttings [7]. Low concentrations of trans-zeatin riboside (ZR) are conducive to the formation and development of adventitious roots [8]. In addition, changes in enzyme activity were found closely related to each stage of the cuttage rooting process. Peroxidase (POD) is involved in auxin metabolism, phenolic substance synthesis, and cell wall lignification. An increase in POD activity during the induction period of root primordium was considered a good indicator of rooting ability. Highly active polyphenol oxidase (PPO) promoted an adventitious root induction and a differentiation of cuttings [9], while the enzyme, indoleacetic acid oxidase (IAAO), oxidized IAA, which was closely related to rooting. A reduced IAAO enzyme activity during the induction period of root primordia of easily rooted tree species was found conducive to the formation of calli from cuttings, while the opposite trend occurred in hard-rooted tree species [10]. The effects of endogenous hormones and enzyme activities on the rooting process of *A. mono* cuttings involving rejuvenation and non-rejuvenation methods have not been reported.

In this study, 0.5 m of *A. mono* shoots were subjected to rejuvenation and non-rejuvenation treatments to make cuttings. Exogenous plant-growth-regulation substances (NAA + IBA) of 1500 mg/L were used to treat cuttings, after which the cuttings were readied. Endogenous hormone levels and enzyme activities in the rooting process of cuttings were measured. The effects of exogenous plant growth regulators on the rooting of rejuvenated and non-rejuvenated cuttings were analyzed and the physiological differences between the rejuvenated and non-rejuvenated cuttings during rooting were compared. The results explained the mechanism of rooting in tree cuttings promoted by rejuvenation measures and provided a scientific basis for the optimization of the cutting propagation method to promote the mass propagation of fine *A. mono* varieties.

## 2. Results

### 2.1. Rooting Characteristics, Rooting Rate, and Callus Rate of A. mono Cuttings

There were three types of cuttage rooting, namely callus rooting, cuticle rooting, and callus and cuticle rooting. Callus rooting accounted for 80% of all rooting types and the number and quality of the rooting were superior compared to those of cuticle rooting. The rooting process was divided into callus formation period (0–10 d), root primordium induction period (10–20 d), adventitious root formation period (20–30 d), and adventitious root extension period (30–50 d). At these time points, the differences between rejuvenation and non-rejuvenation treatment are statistically significant. The rooting rate of non-rejuvenated cuttings treated with NAA + IBA (1:1) mixture of 1500 mg/L was significantly higher compared to that under other treatments (*p* = 0.05) and 33.33% higher compared to those treated with rejuvenation and NAA + IBA. The rooting rate of the rejuvenation control or non-rejuvenation control was significantly lower compared to that of other treatment groups (*p* = 0.05), indicating that exogenous plant-growth-regulator treatment improved the rooting rate. The highest callus rate was 66.67% for non-rejuvenation NAA + IBA treatment, which was significantly higher compared to that of other control and treatment groups (*p* = 0.05). The lowest callus rate of 16.67% was observed for non-rejuvenation control (Table 1).

### 2.2. Changes in IAA Content during the Rooting Process of A. mono Cuttings

In the process of *A. mono* cuttings and rooting, the trend of the endogenous hormone, IAA, in the F-Control (*A. mono* rejuvenation treated without exogenous plant growth regulators) and W-Control (*A. mono* non-rejuvenation treated without exogenous plant growth regulators) was roughly the same, showing an ascending–descending–ascending trend (Figure 1a,b). IAA content increased rapidly during the callus formation stage (0–10 d) but decreased rapidly with the continuous development of root primordia and the expression of adventitious roots. IAA content in the rejuvenated control group reached its peak value at 20 d of culture, which was 10 d earlier compared to the non-rejuvenation control group. IAA content increased gradually in the adventitious root expression and extension stages in the rejuvenation control or non-rejuvenation control groups, which was conducive to the growth and development of adventitious roots.

Endogenous IAA levels in F-NAA + IBA (*A. mono* rejuvenation treated with exogenous plant growth regulators) and W-NAA + IBA (*A. mono* non-rejuvenation treated with exogenous plant growth regulators) showed a similar trend. The decrease in IAA content during callus formation could be due to the early formation of callus caused by hormone treatment, consuming a large amount of IAA. However, the initial IAA content in the rejuvenation treatment group was higher compared to the non-rejuvenation treatment group, reaching the lowest value after 20 d of culture. At the adventitious root formation and elongation stages, IAA content in the rejuvenation treatment group increased first and then decreased significantly, while IAA levels in the non-rejuvenation treatment group increased first and then leveled off. The increase in IAA levels provided favorable conditions for the formation and elongation of adventitious roots.

### 2.3. Changes in ABA Content during the Rooting Process of A. mono Cuttings

Endogenous ABA content in the F-Control and W-Control showed an increasing trend at the callus formation stage (0–10 d) (Figure 2a,b). The first peak value was attained 10 d after treatment. At the root primordium induction stage, ABA content in the rejuvenation control group remained at a relatively stable level, reaching a peak value of 618.40 ng·g^−1^FW after 20 d of cutting, while ABA content in the non-rejuvenation control group decreased rapidly. In the adventitious root formation and elongation stages, changes in ABA content in the rejuvenation control group were slower compared to the non-rejuvenation control group for 10 d.

The initial ABA content in cuttings treated with exogenous plant growth regulators was higher compared to that in the control group. The application of exogenous NAA + IBA increased the ABA content in cuttings to improve resistance. Within 20 d of cutting, the ABA content in the rejuvenation treatment group remained stable, while the ABA content in the non-rejuvenation treatment group showed an increase first, followed by a decrease, which was roughly the same trend as that of the corresponding control group. The ABA content decreased in the adventitious root formation stage (20–30 d), reaching its lowest value (387.37 ng·g^−1^FW and 377.45 ng·g^−1^FW) after 30 d of cutting. This indicated that a lower ABA content was required for adventitious root differentiation. The ABA content in the treatment group increased gradually with the elongation of adventitious roots, and reaching its peak value (667.96 ng.g^−1^FW and 767.08 ng.g^−1^FW) at 40 d.

### 2.4. Changes in GA_3_ Content during the Rooting Process of A. mono Cuttings

The content of GA_3_ increased slightly during callus formation in F-Control and W-Control, improving the resistance of *A. mono* cuttings. During the formation of root primordia and adventitious roots, GA_3_ content decreased gradually, indicating that a low concentration of GA_3_ was required for the formation of root primordia and adventitious roots. In the adventitious root elongation period, GA_3_ content of the two cuttings showed the opposite trend, where the GA_3_ content in the rejuvenation control group gradually decreased and the GA_3_ content in the non-rejuvenation control group gradually increased.

Initial GA_3_ content in the F-NAA + IBA was lower compared to the W-NAA + IBA and the application of exogenous NAA + IBA reduced the rejuvenation treatment group GA_3_ content. In the root primordium induction stage and the early stage of adventitious root differentiation, the formation of adventitious roots consumed more GA_3_, and the GA_3_ content decreased significantly. The GA_3_ content in the rejuvenation treatment group increased rapidly on day 20 after cutting, while that in the non-rejuvenation treatment group showed an increasing trend on day 30 (Figure 3).

### 2.5. Changes in ZR Content during the Rooting Process of A. mono Cuttings

As shown in Figure 4, the initial ZR content in the cuttings of the F-Control was 75.96 ng·g^−1^FW, which was 51.31% higher compared to the initial ZR value in the cuttings of the W-Control. This suggested that a high concentration of ZR was required for the induction and differentiation of root primordium. Within 20 d of cutting, the ZR content in the cuttings of the rejuvenated control group decreased significantly, while the non-rejuvenated control group accumulated a certain amount of ZR before entering the root primordium induction stage. At 20 d of cutting, ZR content in the F-Control and W-Control groups was the lowest (40.97 ng·g^−1^FW and 46.66 ng·g^−1^FW). During the adventitious root elongation stage, the ZR content in the F-Control group decreased first and then increased, while the ZR content in the F-Control group showed an increase throughout, reaching the highest value of 82.70 ng·g^−1^FW on the 50th day.

The ZR content in the F-NAA+IBA first decreased and then increased to maintain a stable level within 0–30 days, while the ZR content in the W-NAA + IBA first increased and then decreased to maintain a stable level. It could be because the rejuvenation treatment group entered the callus formation stage in advance, while the non-rejuvenation treatment group needed to accumulate a certain amount of ZR content. In the adventitious root elongation stage, the ZR content in the rejuvenation treatment group decreased first and then increased, while the ZR content in the non-rejuvenation treatment group decreased throughout, reaching the lowest value of 44.34 ng·g^−1^FW on the 50th day.

### 2.6. Changes in POD Activity during the Rooting Process of A. mono Cuttings

POD activity of the F-Control and W-Control showed an increasing trend during 0–20 d of cutting, indicating that a high concentration of POD was required for the induction and differentiation of root primordium. During the adventitious root formation stage (20–30 d), POD activity decreased in both cuttings, indicating that a low concentration of POD was required for adventitious root formation. In the adventitious root elongation period, POD content in the cuttings of the F-Control and W-Control groups showed different trends. The POD content in the rejuvenation control group first increased and then decreased, while the POD content in the cuttings without rejuvenation showed a constant decrease, which could be due to the rejuvenation measures.

Initial POD activity in the F-NAA + IBA and W-NAA + IBA was higher compared to the control group, while the treatment of exogenous NAA + IBA improved POD activity in the cuttings. In the callus formation stage (0–10 d), POD activity in the cuttings treated with and without rejuvenation increased similarly. The POD content of the rejuvenation treatment group decreased within 10–30 d of cuttings, while that in those not treated with rejuvenation decreased within 20–40 d of cuttings. The induction of root primordia and adventitious roots in rejuvenation treatment group were earlier 10 d compared to non-rejuvenation treatment group. In the adventitious root elongation stage, POD activity showed an increase (Figure 5).

### 2.7. Changes in PPO Activity during the Rooting Process of A. mono Cuttings

The PPO activity of the F-Control showed a decreasing trend within 0–20 d of cutting, while the PPO activity of the W-Control increased first and then decreased, reaching the lowest value of 6.22 U·g^−1^FWmin^−1^ at 20 d of cutting. This demonstrated that a low concentration of PPO was required for the formation of root primordium. In the adventitious root formation stage, PPO activity increased, whereas in the late adventitious root elongation stage, the PPO activity of the non-rejuvenation control group showed a downward trend (Figure 6).

For the F-NAA + IBA and W-NAA + IBA, the initial value of the PPO activity was higher compared to the corresponding control group, indicating that the treatment of exogenous plant growth regulators improved PPO activity. The changing trend of PPO activity of cuttings belonging to the rejuvenation and non-rejuvenation treatment groups was roughly the same, as both showed a decreasing trend at 0–20 d of cutting, reaching the lowest value (10.67 U·g^−1^FWmin^−1^ and 4.45 U·g^−1^FWmin^−1^) at 20 d of cutting. However, the PPO activity of the rejuvenation treatment group was always higher compared to the non-rejuvenation treatment group. This could be a reason for a lower callus rate in the rejuvenation treatment group. The period of adventitious root expression and elongation was 20–40 d of cutting, with a rapid increase in PPO activity in the treatment group, aiding adventitious root elongation. At a later stage of adventitious root elongation, the PPO activity decreased in the rejuvenation treatment group, while it increased in the non-rejuvenation treatment group (Figure 6).

### 2.8. Changes in IAAO Activity during the Rooting Process of A. mono Cuttings

IAAO activity of the F-Control and W-Control increased during the callus formation stage. The IAAO activity of the rejuvenation control group continued to increase at the root primordium induction stage, reaching the peak at 20 d of cutting. However, the IAAO activity of the non-rejuvenation control group decreased in the root primordium induction stage, with an increase in IAA content, conducive to the generation of adventitious roots. In the adventitious root formation stage, IAAO activity in both cuttings was at a low level. In the adventitious root elongation stage, IAA content decreased, while IAAO activity increased slightly, maintaining stability.

The trend of IAAO activity in cuttings treated with NAA + IBA was significantly different. The IAAO activity decreased slightly in the rejuvenation treatment group at the callus formation stage, whereas it increased significantly in the non-rejuvenation treatment group. During the root primordium induction period, the IAAO activity in the group without rejuvenation treatment decreased rapidly and was lower compared to the group with rejuvenation treatment (Figure 7).

## 3. Discussion

### 3.1. Endogenous Hormone IAA Could Promote the Formation of Adventitious Roots in the Cuttings of A. mono

Endogenous hormones play an important role in regulating plant growth and development and promoting the generation of adventitious roots from cuttings. Various hormones interact with each other to regulate rooting [11,12], mainly acting on receptor proteins to promote the division and growth of root primitive cells by transporting sugars and other substances [13]. IAA is one such main hormone inducing the formation of adventitious roots. IAA can significantly induce the initiation of root primordia and increase the number of root primordia [14]. The results of our study showed that the IAA content in the cuttings treated with exogenous hormones decreased during the induction of adventitious roots, which could be due to the effect of exogenous hormones on the formation of callus, thereby consuming a large amount of IAA in advance, while the control group required an accumulation process before callus formation. Studies found that a high concentration of IAA increased the number of root meristem cell divisions, while a low concentration of IAA accelerated the differentiation of cells in the elongation region of the root system [15]. In this study, the IAA content of the control group increased rapidly during the induction and differentiation stages of adventitious roots, which was conducive to the differentiation of root primordia into adventitious roots [16]. Shang C Q et al. found that the IAA had a rapid transit peak at 6 h after planting (hAP) and declined thereafter. These findings demonstrate the positive effect of auxin on AR induction [17]. Hence, IAA was considered an important endogenous hormone to promote the formation of adventitious roots in the cuttings of *A. mono* [13].

### 3.2. Endogenous Hormone ABA on Adventitious Roots in the Cuttings of A. mono

In the cutting process of *A. mono*, ABA is a hormone that can inhibit the rooting of cuttings. A decrease in ABA content aided the formation of root primordia and new roots during the early stage of cuttings [18], returning to its normal level at a later stage. The results of this study showed that ABA content in cuttings increased slightly at the early stage of cuttings, which could be a protective mechanism to deal with adversities initiated after the cuttings were removed from the mother plant, thus reducing injury to the wound, while abiotic stress response enhanced the stress resistance of plants [19,20]. The ABA content in NAA + IBA-treated cuttings reached the lowest level at 30 d, while that of the control group reached the same at 20 d of cutting, which then gradually increased. These results indicated that exogenous hormones could effectively reduce ABA content in cuttings at the adventitious root induction stage, thereby increasing the rooting rate of cuttings. In the adventitious root elongation period, ABA content again decreased, which was consistent with studies on cucumber (*Cucumis sativus*), where a low ABA concentration promoted adventitious root elongation [21].

### 3.3. Endogenous Hormone GA_3_ Have Different Effects of Adventitious Roots in the Cuttings of A. mono

GA_3_ is a hormone produced in plants that affects plant growth and development. There are different opinions on the effect of GA_3_ on cuttings. On the one hand, GA_3_ shows the opposite effect of auxins, inhibiting the formation of adventitious roots by inhibiting the division of root primordium or preventing the formation of root primordia induced by auxins [7]. On the other hand, it is believed that GA_3_ plays different roles at different rooting development stages, with low concentrations of GA_3_ cooperating with auxins to induce adventitious roots, conducive to the induction and differentiation of root primordia [22]. This study found that the GA_3_ content increased during callus formation, inhibiting adventitious root formation, while GA_3_ content in the non-rejuvenated treatment group decreased significantly during root primordium induction and adventitious root formation stages. This could be a reason for a higher rooting rate in the non-rejuvenated group compared to the rejuvenated group. Regarding the mechanism of the effect of gibberellin on rooting, studies showed that gibberellin does not affect the IAA signaling pathway nor its upstream genes, but that it inhibits adventitious root formation in both *Populus* L. and *Arabidopsis thaliana* by affecting auxin transport [23].

### 3.4. Endogenous Hormone ZR Conducive to the Growth and Development of Adventitious Roots of A. mono

ZR is the main transport form of cytokinins in woody plants, promoting cell division and differentiation and influencing adventitious root formation [24,25]. Some studies showed that ZR inhibited plant rooting during the rooting process of cuttings, where a low content of ZR was conducive to root primordium differentiation and a high content promoted the formation of adventitious roots [26]. In this study, ZR content decreased significantly at the root primordium induction stage, which could be related to the consumption of ZR during the differentiation process of root primordium. This was consistent with the test results of Geng Xinlu on *Eucommia ulmoides* [27]. Meanwhile, ZR content increased during the adventitious root formation stage of *A. mono*, which was conducive to the growth and development of adventitious roots. This was consistent with the results of Shao Fengxia’s study on *Zizyphus jujuba* [6].

### 3.5. POD Is Closely Related to the Induction and Growth of Adventitious Roots of A. mono

POD is closely related to the induction and growth of adventitious roots and is also involved in auxin metabolism, respiration, photosynthesis, etc. It plays an important role in cell division and differentiation, the formation and growth of root primordium, and adaptation to adverse environments [28]. POD is closely related to rooting induction and the expression of cuttings [9]. Studies showed that a decrease in POD activity was beneficial for the induction of root primordium, which was observed similarly in the rooting process of rejuvenated *A. mono* cuttings in our study [18,29]. During the early stages of cutting, the cuttings perceive injury signals and increase POD activity, which not only responds to stress, but also removes free radicals and other substances from the cuttings. Hence, it was believed that a high level of POD activity before cutting rooting was beneficial to rooting [30]. However, the *A. mono* cuttings without rejuvenation treatment showed different trends during the adventitious root elongation period. This demonstrated that the effect of POD activity on callus formation and adventitious root formation of *A. mono* with rejuvenation treatment should be investigated further.

### 3.6. PPO Could Promote the Elongation of Adventitious Roots of A. mono

PPO is a copper-containing oxidase. In the early cutting stages of *A. mono*, PPO activity increased rapidly and oxidized phenolic substances into quinones [31], promoting the formation of calli and adventitious roots. Opinions differ on whether PPO activity is necessary for the rooting process of cuttings. Some studies believed that rooting performance was not necessarily related to PPO activity levels [32]. In this study, PPO activity decreased in the early stage of cuttings, which could be due to the separation from the mother plant and to reduce damage to cuttings. PPO activity increased in the induction and growth stages of adventitious roots. PPO also catalyzed the synthesis of phenolic substances into rooting cofactors, which were beneficial for the formation of callus. In the adventitious root elongation period, phenolic substances were decomposed and PPO activity gradually decreased, promoting the elongation of adventitious roots and providing endogenous hormones for normal seedling growth.

### 3.7. IAAO Could Control IAA Concentration and Promote the Growth and Development of Cuttings of A. mono

IAAO degrades IAA to control IAA concentration and promote the growth and development of cuttings at different stages. Studies showed that during the adventitious root elongation period, IAAO activity gradually increased and a low concentration of IAA promoted adventitious root elongation by oxidizing more IAA [33]. In Nag’s test, with the formation of a large number of adventitious roots, the IAAO activity of the K-IBA-treated group gradually increased, resulting in a decrease in the IAA content in vivo. This promoted the growth of adventitious roots [34]. In this study, the changing trend of the IAAO activity and IAA content in cuttings treated with exogenous hormones was roughly the opposite. A study on tea trees found that the IAAO activity was highly consistent with changes in the IAA content, showing a reciprocal relationship [9].

## 4. Materials and Methods

### 4.1. Plant Materials

*A. mono* were obtained from the Maoershan Experimental Forest Farm of the Northeast Forestry University, Heilongjiang, China (127°30′–127°34′ E, 45°21′–45°25′ N). They were seven-year-old healthy artificial seedlings without pests or diseases. In May 2021, half of the *A. mono* were randomly selected and rejuvenation was achieved by a stumping treatment of 0.5 m. The rejuvenation treatment of *A. mono* grows young branches 0.5 m above the ground for cutting propagation.

Source of cutting matrix material: Perlite was obtained from Houpu Mining Material Factory, Henan, China and the peat soil used was the Pind Strup brand matrix (Denmark) from Pindstrup Horticulture (Shanghai) Co., Ltd., Shanghai, China. Before cutting, the substrate was soaked in a potassium permanganate (KMnO_4_) solution with a mass concentration of 0.5% for 24 h and then cleaned with water.

### 4.2. Cutting Bed and Test Conditions

The experiment was conducted in the greenhouse of the Institute of Flower Bioengineering, Northeast Forestry University, China. The cutting bed was 20 m long and 1.5 m wide and the matrix was filled with black seedling trays (37 cm long, 30 cm wide, 7.5 cm high) with holes for leakage at the bottom. A curved plastic shed with a height of 0.8 m was built on the seedling bed. Automatic spraying facilities were used inside the plastic shed (spraying for 2 min every 2 h during the day and 2 min every 4 h at night). The temperature was maintained at 26–31 °C and humidity at 70–80%. The plastic shed was covered with a movable 70% black shade net to regulate light intensity. The temperature inside the plastic shed was controlled by ventilation at both ends of the shed and the greenhouse cooling system.

### 4.3. Collection and Production of Cuttings

In July 2022, the middle and lower branches of the mother plants of *A. mono* rejuvenation and non-rejuvenation stock plant was collected and transported in an incubator to the greenhouse of the Northeast Forestry University in time. They were then wetted with cold water and stored in a cool place for later use. The semi-lignified part of the branches was split into cuttings of 6–10 cm and disinfected with 1000 times carbendazim. The base leaves of the cuttings were removed and the top two half leaves were kept and cut into horseshoe shapes 1 cm away from the bottom bud of the cuttings for use when required.

A single-factor completely randomized block experimental design was used. The base of the rejuvenated and non-rejuvenated cuttings was treated with a NAA + IBA (1:1) mixture having a mass concentration of 1500 mg/L. After standing for 2 min, the cuttings were transferred to a disinfected perlite matrix. Water treatment was used as the control.

During the cutting process, a stick having a thickness similar to the cuttings was used to drill holes into the matrix (depth of 2–3 cm). After cutting, the cuttings were in close contact with perlite, with the leaves of the cuttings remaining in the same direction. After the cutting process was completed, automatic spraying was initiated to keep the leaf surface of the cuttings moist. The substrate and cuttings were sprayed with 1000 times carbendazim every week for disinfection.

### 4.4. Index Observation

There were four replicates in each treatment and 24 cuttings in each replicate. The cuttings were distributed as 6 × 4 cuttings in each seedling tray, with samples being taken at 0, 20, 30, 40, and 50 days (d) of cutting. Four cuttings were randomly selected from each plate during sampling and a total of sixteen cuttings were taken from each treatment (4 plates × 4 cuttings). The cuttings were washed with water, dried, and immediately placed in an ice box and brought to the laboratory. The characteristics of the base of the cuttings were observed and recorded, after which, the phloem of 2–3 cm and a small amount of xylem at the base of the cuttings were quickly peeled off. The samples were cut, mixed well, wrapped tightly with tin foil, frozen with liquid nitrogen, and stored in an ultra-low temperature refrigerator (−80 °C) for further use.

### 4.5. Measurement Index and Data Analysis

Determination of endogenous hormones: The contents of IAA, ABA, ZR, and GA_3_ in *A. mono* cuttings were determined by enzyme-linked immunoassay (ELISA) at the Shanghai Enzyme Linked Biotechnology Co., Ltd. (120°52′–122°12′ E, 30°40′–31°53′ N) and the experiments for each index were repeated thrice.

Enzyme activity determination: The activities of POD, PPO, and IAAO were determined spectrophotometrically using the kits of China Suzhou Grisi Biotechnology Co., LTD (119°55′–121°20′ E, 30°47′–32°02′ N). Reagents in the kits were all configured reagents, convenient for the rapid determination of enzyme activity, which was measured according to the manufacturer’s instructions. Experiments for each index was repeated thrice.

Microsoft Excel 2019 was used for statistical data analysis. SPSS v.19.0 was used for one-way analysis of variance (ANOVA) and Duncan’s multiple range test (DMRT) (alpha = 0.05, alpha = 0.01). Sigmaplot was used (2011, v.12.5, SYSTAT, US) for graphing.

## 5. Conclusions

Exogenous growth-regulating substances significantly improved the rooting rate of *A. mono*. The rooting rates of rejuvenated (29.17%) and non-rejuvenated (62.5%) cuttings were 7.25 and 15.63 times higher compared to the control (4.17%), respectively. Exogenous hormones (NAA + IBA) increased the contents of IAA and ABA and the activities of POD and PPO enzymes in the cuttings of *A. mono*. The lower contents of IAA, ABA, and PPO enzymes were beneficial for the induction of root primordia, while a higher content of these enzymes promoted the formation and elongation of adventitious roots. Rejuvenation treatment prolonged the time of increase in ABA content and IAAO activity at the root primordium induction stage, increased ZR content, and decreased POD enzyme activity in cuttings, which could be the reason for the rooting rate in the rejuvenated group being lower compared to the non-rejuvenated cuttings. These results demonstrate that *A. mono* cuttings can achieve the purpose of improving the rooting rate by adding the exogenous hormone (NAA + IBA), which is closely related to the changes of endogenous hormone content and enzyme activity, and these changes of *A. mono* rejuvenation cuttings are different from non-rejuvenation cuttings. Overall, our findings provide a theoretical basis for the further development of technical methods to improve the rooting rate of rejuvenated cuttings.

## Figures and Tables

**Figure 1 ijms-24-11883-f001:**
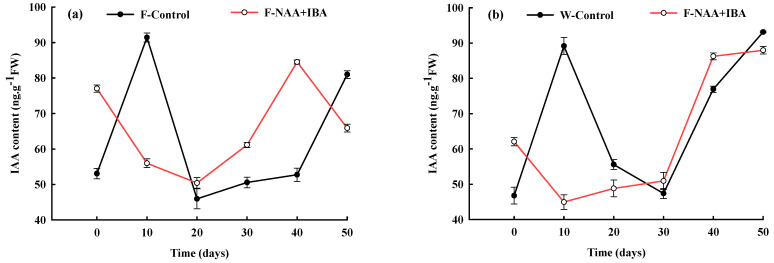
Changes in IAA content during the rooting process of *A. mono* twig cuttings with (**a**) or without (**b**) rejuvenation treatment.

**Figure 2 ijms-24-11883-f002:**
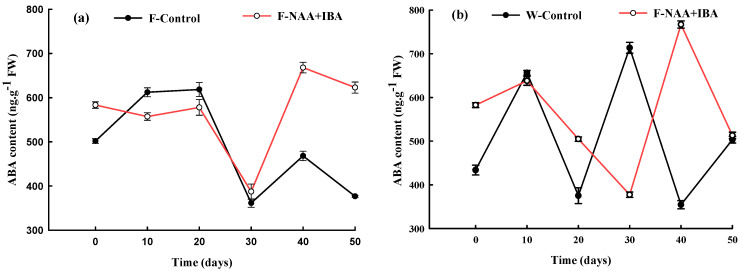
Changes in ABA content during the rooting process of *A. mono* twig cuttings with (**a**) or without (**b**) rejuvenation treatment.

**Figure 3 ijms-24-11883-f003:**
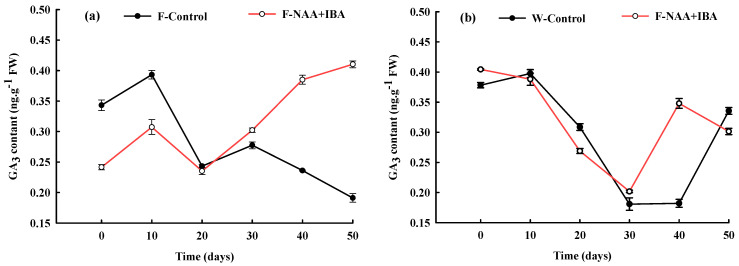
Changes in GA_3_ content during the rooting process of *A. mono* twig cuttings with (**a**) or without (**b**) rejuvenation treatment.

**Figure 4 ijms-24-11883-f004:**
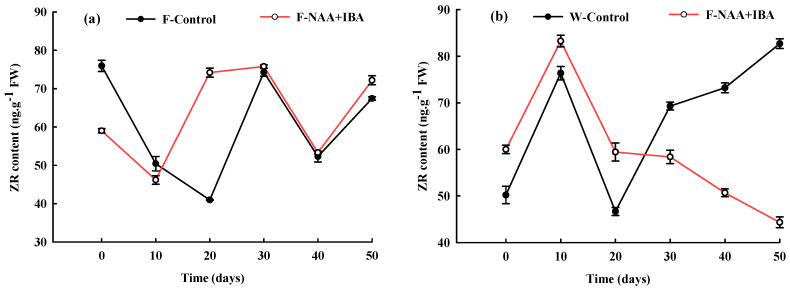
Changes in ZR content during the rooting process of *A. mono* twig cuttings with (**a**) or without (**b**) rejuvenation treatment.

**Figure 5 ijms-24-11883-f005:**
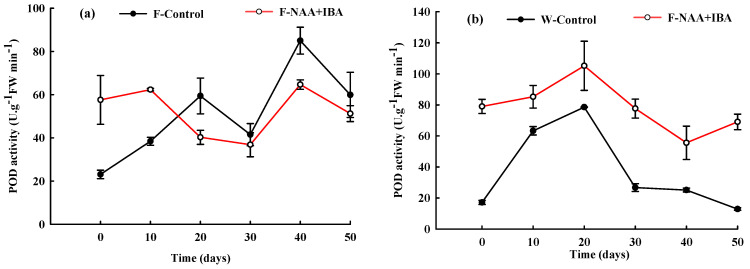
Changes in POD activity during the rooting process of *A. mono* twig cuttings with (**a**) or without (**b**) rejuvenation treatment.

**Figure 6 ijms-24-11883-f006:**
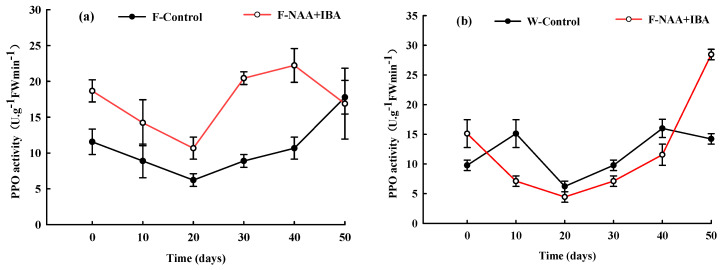
Changes in PPO activity during the rooting process of *A. mono* twig cuttings with (**a**) or without (**b**) rejuvenation treatment.

**Figure 7 ijms-24-11883-f007:**
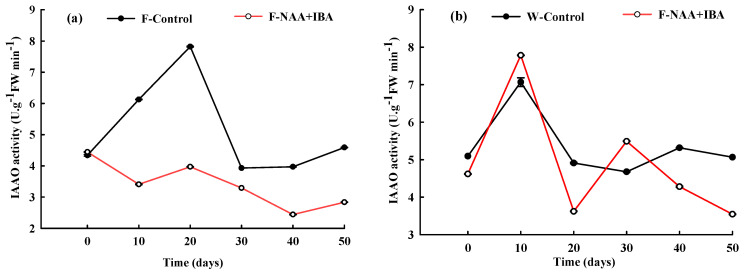
Changes in IAAO activity during the rooting process of *A. mono* twig cuttings with (**a**) or without (**b**) rejuvenation treatment.

**Table 1 ijms-24-11883-t001:** Comparison between rooting rate and callus rate in the cutting process of *A. mono*.

Treatment	Rooting Rate (%)	Callus Rate (%)
Rejuvenation Control	4.17 ± 4.17 c	25.00 ± 4.81 b
Rejuvenation NAA + IBA	29.17 ± 4.17 b	37.50 ± 7.98 b
Non-Rejuvenation Control	4.17 ± 4.17 c	16.67 ± 6.80 b
Non-Rejuvenation NAA + IBA	62.50 ± 4.17 a	66.67 ± 6.80 a

Note: Values are mean ± standard error (SE); the same letter between columns indicates no significant difference between treatments, while different letters indicate a significant difference (*p* ≤ 0.05).

## Data Availability

Not applicable.

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
