# Peer review of "Effect of Exogenous Plant Growth Regulators and Rejuvenation Measures on the Endogenous Hormone and Enzyme Activity Responses of Acer mono Maxim in Cuttage Rooting"

_ijms, 2023, doi:10.3390/ijms241511883_

Round 1

Reviewer 1 Report

The topic is important and interesting. Dut the MS should be improved. For example:

1) The abstract does not reflect the purpose and object of the study; it does not contain the main results of the study. It is necessary to redo the annotation completely.

2) The process of rejuvenation of cuttings is not very clearly written. Was rejuvenation achieved by grafting? Were the plants from the Maoershan Experimental Forest Farm the graft? Write, please, more clearly the process of rejuvenation.

3) Could you decipher the captions: “D-Ck “ and “W-CK” etc?

Author Response

We are very grateful to you for taking the time to read and modify our article again. We find that your comments play a very important role in improving the quality of our papers. We have carefully revised the paper in light of your comments, and please find our response to the comments made below. We marked the modified part of the manuscript in red.

Thank you for considering our revised manuscript!

Point 1: The abstract does not reflect the purpose and object of the study; it does not contain the main results of the study. It is necessary to redo the annotation completely.

Response 1: Thank you very much for your suggestion. We read the abstract carefully and accepted your suggestions. For details please see the abstract.

Point 2: The process of rejuvenation of cuttings is not very clearly written. Was rejuvenation achieved by grafting? Were the plants from the Maoershan Experimental Forest Farm the graft? Write, please, more clearly the process of rejuvenation.

Response 2: Thank you very much for your suggestion. We read the manuscript carefully and accepted your suggestions. For details please see the text (line 343-348).

Point 3: Could you decipher the captions: “F-CK” and “W-CK” etc?

Response 3: Thank you very much for your suggestion. We read the manuscript carefully and accepted your suggestions. For details please see the text (line 101-103, 112-114).

Reviewer 2 Report

I recommend for the authors to work on with the following problems of this manuscript:

1. The use of terms seems to be quite inconsistent. Several examples:

    a) In Abstract, it is stated that "the cuttings were rooted through two pathways, i.e., callus and bark" (lines 18-19). In Results, however, it is stated that "there were three types of cuttage rooting, namely, callus rooting, cuticle rooting, and callus and cuticle rooting" (lines 89-90).

    b) In lines 108-109, the term "pruning" is used instead of "rejuvenation".

    c) Acer mono is called "maple" in line 44, but "sycamore" in line 48.

2. The choice of a relatively high concentration of exogenous auxins - 1500 mg/l NAA and IBA mixture (1:1) - is quite interesting. It is stated in Introduction that the same combination was used by Zhang et al. (lines 53-58; reference No. 5); however, I found that Zhang et al. (2017), in this referenced work, used concentrations of auxins IAA and NAA from 100 to 700 mg/l. Also, Zhang et al. (2017) investigated Malus hupehensis and not Acer mono (although the authors of this manuscript claim otherwise in lines 53-55).

3. The use of the abbreviation "CK" for the "control" is not the best choice in this context, because this abbreviation - "CK" - is usually used for the term "cytokinin" in plant hormone-related literature.

4. The figures, from 1 to 7, are a little bit problematic because of these issues:

    a) It is not explained what the letters "F" and "W" mean in each of the figure.

    b) All the figures have a kind of double title. The expression "during softwood cutting of A. mono" in the main titles is not clear. In my opinion, it would be better to form the titles for the figures in the following manner: "Changes in ... during the rooting process of Acer mono twig cuttings with (a) or without (b) rejuvenation treatment".

    c) It is not indicated at which time points the differences between two treatments are statistically significant. 

5. One of the biggest issues with this manuscript is not explaining clearly in the Materials and Methods section what is meant by "rejuvenation". How "rejuvenated" cuttings were treated (or developed) differently in comparison to "non-rejuvenated" cuttings. How was "rejuvenation" treatment actually applied?

Not bad, but still there are spelling errors that must be checked over.

Author Response

Dear Reviewer,

We are very grateful to you for taking the time to read and modify our article again. We find that your comments play a very important role in improving the quality of our papers. We have carefully revised the paper in light of your comments, and please find our response to the comments made below. We marked the modified part of the manuscript in green.

Point 1: The use of terms seems to be quite inconsistent. Several examples:

  1.   a) In Abstract, it is stated that "the cuttings were rooted through two pathways, i.e., callus and bark" (lines 18-19). In Results, however, it is stated that "there were three types of cuttage rooting, namely, callus rooting, cuticle rooting, and callus and cuticle rooting" (lines 89-90).
  2.   b) In lines 108-109, the term "pruning" is used instead of "rejuvenation".
  3. c) Acer mono is called "maple" in line 44, but "sycamore" in line 48.

Response 1: Thank you very much for your suggestion. We read the manuscript carefully and accepted your suggestions. For details please see the text (line81-82, 102-103, 41).

Point 2: The choice of a relatively high concentration of exogenous auxins - 1500 mg/l NAA and IBA mixture (1:1) - is quite interesting. It is stated in Introduction that the same combination was used by Zhang et al. (lines 53-58; reference No. 5); however, I found that Zhang et al. (2017), in this referenced work, used concentrations of auxins IAA and NAA from 100 to 700 mg/l. Also, Zhang et al. (2017) investigated Malus hupehensis and not Acer mono (although the authors of this manuscript claim otherwise in lines 53-55).

Response 2: Thank you very much for your suggestion. We read the manuscript carefully and accepted your suggestions. For details please see the text (line46-48, 443-444).

Point 3: The use of the abbreviation "CK" for the "control" is not the best choice in this context, because this abbreviation - "CK" - is usually used for the term "cytokinin" in plant hormone-related literature.

Response 3: Thank you very much for your suggestion. We read the manuscript carefully and accepted your suggestions. For details please see the figure 1-7 and the text.

Point 4: The figures, from 1 to 7, are a little bit problematic because of these issues:

  1.   a) It is not explained what the letters "F" and "W" mean in each of the figure.
  2.   b) All the figures have a kind of double title. The expression "during softwood cutting of  mono" in the main titles is not clear. In my opinion, it would be better to form the titles for the figures in the following manner: "Changes in … during the rooting process of Acer monotwig cuttings with (a) or without (b) rejuvenation treatment".
  3. c) It is not indicated at which time points the differences between two treatments are statistically significant. 

Response 4: Thank you very much for your suggestion. We read the manuscript carefully and accepted your suggestions. For details please see the figure 1- 7 and the text (line101-103, 112-114, 83-87).

Point 5: One of the biggest issues with this manuscript is not explaining clearly in the Materials and Methods section what is meant by "rejuvenation". How "rejuvenated" cuttings were treated (or developed) differently in comparison to "non-rejuvenated" cuttings. How was "rejuvenation" treatment actually applied?

Response 5: Thank you very much for your suggestion. We read the manuscript carefully and accepted your suggestions. For details please see the text (line343-348, 366-368).

Round 2

Reviewer 1 Report

Authors have revised the abstract, it has become much better. However, I think it's not enough just a listing of the results, but it is necessary to add a conclusion on how changes in the hormonal balance increased the rooting of the cuttings.

Author Response

Response to Reviewer Comments

We are very grateful to you for taking the time to read and modify our article again. We find that your comments play a very important role in improving the quality of our papers. We have carefully revised the paper in light of your comments, and please find our response to the comments made below. We marked the modified part of the manuscript in red and highlight yellow.

Thank you for considering our revised manuscript!

Point: Authors have revised the abstract, it has become much better. However, I think it's not enough just a listing of the results, but it is necessary to add a conclusion on how changes in the hormonal balance increased the rooting of the cuttings.

Response: Thank you very much for your suggestion. We read the abstract carefully and accepted your suggestions. For details please see the abstract and conclusions (line 25-29,422-426).